# The Val158Met Polymorphism in 8-Year-Old Boys and Girls Moderates the Influence of Parenting Styles on Proactive Aggression: Testing the Sensitivity to the Environment

**DOI:** 10.3390/brainsci13111513

**Published:** 2023-10-26

**Authors:** Nora del Puerto-Golzarri, Aitziber Azurmendi, José Manuel Muñoz, María Rosario Carreras, Eider Pascual-Sagastizabal

**Affiliations:** 1Department of Basic Psychological Processes and Their Development, Faculty of Psychology, University of the Basque Country (UPV/EHU), 20018 San Sebastián, Spain; nora.delpuerto@ehu.eus (N.d.P.-G.); eider.pascual@ehu.eus (E.P.-S.); 2Psychology Department, Faculty of Education Sciences, University of Cadiz (UCA), 11519 Puerto Real, Spain; josemanuel.munoz@gm.uca.es (J.M.M.); rosario.carreras@gm.uca.es (M.R.C.)

**Keywords:** aggressive behavior, parenting styles, COMT, differential susceptibility, diathesis-stress

## Abstract

The aim of the study was to explore the possible vulnerability (diathesis-stress), susceptibility (differential susceptibility), or vantage (vantage sensitivity) properties of COMT gen Val158Met polymorphism to adverse and favorable parenting styles from both parents in relation to children’s reactive and proactive aggressive behavior. Within 279 eight-year-old children (125 girls and 154 boys) from Spain, reactive and proactive aggressive behavior was measured through the “Reactive and Proactive Questionnaire” (RPQ). Saliva samples were collected to genotype for the COMT Val158Met polymorphism via real-time PCR. Finally, parenting styles were assessed using the “Parenting Styles and Dimensions Questionnaire” (PSDQ). The results revealed that for boys, the Met allele was a vulnerability factor for proactive aggression in response to low-authoritative parenting from the father. For girls, it was the Val allele, the vulnerability variable to the high authoritarian style of the father, and the susceptibility factor to the authoritative style of the mother over proactive aggression. The results are discussed, considering possible sex differences. Our results indicate that the COMT Val158Met polymorphism is a biological variable that confers greater sensitivity to the environment.

## 1. Introduction

Aggressive children and adolescents are more likely to experience mental health problems [1,2], as well as other detrimental long-term consequences, including school difficulties, unemployment, divorce, partner abuse, and neglectful and abusive parenting [3,4,5]. They are also at risk of becoming violent adolescents and adults [6]. Consequently, aggressive behavior is considered an obstacle or impediment to social consolidation [7] and is regarded as one of the most concerning social and health problems nowadays [8]. However, from an evolutionary perspective, aggressive behavior is considered to be adaptive and derived from strategies shaped, in general, by natural selection [9,10] and sexual selection in particular [9]. The latter process molds sex differences in the expression of aggression in both males and females as it acts as a tool for ensuring survival [11,12,13] and its expression depends on environmental factors [14]. Therefore, aggressive behavior with a phylogenetic basis inherent to human beings, involving a wide range of physical, cognitive, emotional, and social factors [15]. Based on the adaptive function of this behavior, its most common classification distinguishes between reactive and proactive aggression [16]. Proactive aggression is commonly described as aggressive behavior aimed at harming or injuring another person to gain a benefit [17]. Reactive aggression comprises a set of aggressive actions carried out in response to a stimulus perceived as provocative or threatening [17]. As it has been observed with other types of aggression, sex differences have also been found in this classification, with boys exhibiting higher levels of both reactive and proactive aggression [18,19,20].

Therefore, human development, including aggressive behavior, is influenced by different aspects of the physical and social context. However, individuals differ in their degree of sensitivity to the same environmental conditions [21]. These individual differences have traditionally been analyzed using the diathesis-stress model [22,23], which suggests that certain individuals, due to their endogenous characteristics, are disproportionately vulnerable to the negative effects of adverse experiences or environments compared to their less vulnerable counterparts. In line with an evolutionary view, another theoretical model called the differential susceptibility model [24,25,26] proposes that certain individuals, based on their individual characteristics, are more susceptible to both negative and positive environmental influences. They benefit more from positive environments while also being more vulnerable to negative environments. Lastly, a more recent theory known as the vantage sensitivity theory [24] suggests that some individuals, due to their endogenous factors, are more sensitive and respond more favorably to positive environmental conditions, presenting an advantage over those who do not have this response capacity. 

Aggression is recognized as a diverse and multidimensional behavior [16,17], which as various studies [27] and a meta-analysis [28] have indicated, has a heritability of approximately 40–50% for the risk of aggressive behavior, including both reactive and proactive aggressive behavior [29]. However, it is worth noting that differences in heritability variance have been found, with males having a higher heritability than females, as found in the study by [7]. Additionally, the effect of heritability also changes over time, with genetic factors becoming more predominant in adulthood than in childhood [7]. Based on this, some research has explored the role of several genetic polymorphisms as factors of vulnerability, susceptibility, or advantage in response to the environment. Since pathological aggression could be related to the reduction of the neural reward system, Chen et al. [30] It was hypothesized that genes involved in the dopaminergic system could be relevant in the etiology of aggression. Among the different genes involved in the dopaminergic system, one of the most studied genes is the COMT gene. This gene contains a single nucleotide polymorphism (Val158Met) that results in the substitution of the amino acid valine (Val) by methionine (Met) in the enzyme COMT [31]. This substitution is functionally significant as the homozygous genotype for the Met158 allele exhibits three- to four-fold decreased enzymatic activity when compared to the homozygous genotype for the Val158 allele [31]. As a result, the Met allele leads to reduced degradation of catecholamines, resulting in higher concentrations in the prefrontal cortex [32,33]. Thus, the Met allele, with lower enzymatic activity, has been associated with aggressive behavior in children and adolescents [34]. It has also been linked to psychiatric abnormalities that predispose to aggression in patients with borderline personality disorder [35,36]. However, the Val allele has also been related to aggressive behavior in studies conducted on children diagnosed with attention deficit hyperactivity disorder [37,38] and in a meta-analysis [27]. Additionally, the initial study by Kuperman et al. [39], who first proposed the role of the COMT enzyme in aggression, found an inverse correlation between COMT activity and hostility, as well as a positive correlation with impulsivity. This suggests potential physiological differences among different forms of aggressive behavior.

When considering the environmental factors that can make children vulnerable, susceptible, or “vantageous”, parenting style emerges as a fundamental characteristic of the family environment. Numerous studies have shown that parenting style contributes to a child’s competence, development, and the development of psychopathology [40]. Parenting styles have been specifically studied in relation to aggressive behavior, and the association has been found to vary depending on the parenting style used [41]. Based on the interaction of the dimensions of sensitivity and demand, Baumrind [42,43] identified three types of parenting styles, namely, authoritative, authoritarian, and permissive. On one hand, the authoritative parenting style has been observed as a predictor of reduced externalizing problems [44,45] and aggression [46]. On the other hand, the authoritarian parenting style inhibits prosocial behavior [47] and predicts aggression in children [48,49,50,51]. Finally, the permissive parenting style has been associated with lower levels of prosocial behavior [52] and positively linked to externalizing problems [39] and aggressive behavior in children and adolescents [53,54].

Taking into account the models described, several studies have tested the role of the COMT polymorphism as a potential vulnerability, susceptibility, or vantage factor to the environment over aggression. One such study conducted by Hygen et al. [55] provided evidence supporting the role of the COMT polymorphism. They observed that children homozygous for the Val allele exhibited higher levels of aggression compared to carriers of the Met allele when exposed to significant life events. However, in the absence of such events, Val allele carriers exhibited lower levels of aggression compared to individuals carrying the Met allele, which is consistent with the differential susceptibility hypothesis. Another study by Tuvblad et al. [56], also interpreted within the context of the differential susceptibility theory, found that children homozygous for the Val allele demonstrated higher levels of physical aggression than Met allele carriers when exposed to high levels of violence. However, when these same children were exposed to positive parent–child relationships, they exhibited lower levels of aggression compared to Met allele carriers.

In contrast to the previous studies but yet consistent with the differential susceptibility theory, Zhang et al. [57] discovered that adolescents carrying the Met allele displayed higher levels of reactive aggression when exposed to low levels of positive parenting. However, they also exhibited lower levels of reactive aggression when exposed to positive parenting. There is additional research that appears to present contradictory findings. For instance, Wang et al. [58] found that both Val allele homozygotes and Met allele carriers showed differences in feelings of hostility and aggressive motivation based on whether they experienced social exclusion or inclusion. However, the differences were more pronounced among individuals carrying the Met allele. However, this research is not the only study in which both the Val and the Met alleles have been found to confer sensitivity to the environment. Sulik et al. [59] reported that the Met/Val and Val/Val genotypes in girls, as well as Met/Met genotype in boys, moderated the relationship between internalizing symptoms and an unsupportive parental context in line with the diathesis-stress theory. Additionally, these genotypes moderated the relationship between a supportive parental context and inhibitory control, which aligns with the vantage sensitivity theory. 

Considering the inconsistent findings reported by previous research and the lack of studies examining the role of the COMT polymorphism as a possible susceptibility, vulnerability, or vantage factor to the environment over reactive and proactive aggression, the present study aims to investigate whether or not the Val158Met polymorphism moderates the relationship between parenting style and reactive and proactive aggression among children, taking into account possible sex differences in aggressive behavior. The study will be guided by the diathesis-stress, differential susceptibility, or vantage sensitivity theories. Thus, we hypothesized that the presence of the Met158 allele in an adverse family context would be associated with higher levels of aggressive behavior, while in a favorable context, it would predict less aggressive behavior compared to nonsusceptible ones, supporting the differential susceptibility theory. Furthermore, acknowledging that reactive and proactive aggressive behavior are influenced by various environmental and genetic factors [18] and recognizing the limited understanding of the biology of proactive aggression [60], the study will also explore the potential association between the environment, the COMT polymorphism, and the different forms of aggression.

## 2. Materials and Methods

### 2.1. Participants

The sample group comprised 279 8-year-old children (125 girls and 154 boys) from Spain. Diverse authors have proposed that the shift from early to middle childhood represents a “critical phase” crucial for shaping specific behavioral strategies. This significance arises from the vulnerability of children to environmental influences during this period and the potential impact of biology on their adaptability [61,62]. To obtain this sample, several schools were contacted in the city where the researchers’ university is located, and seven of them gave their written consent. Afterwards, the families whose socioeconomic status was medium-high and high were contacted, and a total of 279 families gave their written consent for their children to participate in the investigation. The project was approved by the Ethics Committee of the institution to which the authors belong, and the procedure complied with national legislation.

### 2.2. Instruments

Reactive and proactive aggression was measured using the Spanish version, adapted by Andreu et al. [63], of the ‘Reactive-Proactive Aggression Questionnaire’ (RPQ), which was originally developed by Raine et al. [17]. Although it was directed towards adolescents, it was also originally tested on 7-year-old children [17], and it was used in children in the same age range as our sample [64,65]. This self-report test comprises 23 items rated on a three-point Likert-type scale (0 = never, 1 = sometimes, and 2 = often) to test the proactive and reactive forms of aggression. The reactive aggression scale was composed of 11 items and had a reliability of α = 0.75 in our sample, while proactive aggression is made up of 12 items, and the reliability obtained in our sample was α = 0.86.

The Parenting Styles and Dimensions Questionnaire (PSDQ) [66] was used to measure authoritative, authoritarian, and permissive parenting styles, and it was answered by both the mother and the father. It contains 62 items on a 4-point Likert-type scale (1 = never, 2 = occasionally, 3 = very often, and 4 = always) to measure the frequency with which parents spend doing behaviors related to their children to get a score for each parenting style. The first scale, the authoritative one, consisted of 27 items and had a reliability of α = 0.89 in our sample. The second scale, authoritarian style, was measured through 20 items, and the reliability of the scale was α = 0.74 in our sample. Finally, the permissive parenting style scale was composed of 15 items and had a reliability of α = 0.54 in our investigation. Due to the low reliability of the permissive scale and following what has been stated by different authors [67,68] that an acceptable score should be higher than α = 0.70, we decided not to include this scale in the analyses.

As for the genetic polymorphism, saliva samples were collected and then processed using a commercial saliva DNA isolation kit (Norgen Biotek Corporation, Thorold, ON, Canada) and then stored at −20 °C until later analysis. Then, to analyze the SNP rs4680 of the COMT gene, we used a pre-designed TaqMan SNP genotyping assay (Thermo Fisher, Waltham, MA, USA) (assay ID: C_25746809_50) that contained fluorescently labeled (VIC (a) and FAM (g) fluorophores) minor groove binders (MGB) probes together with the primers. The real-time PCR was performed using the TaqPath ProAmp Master Mix (Thermo Fisher, Waltham, MA, USA) in a final volume of 10 ul containing up to 20 ng of genomic DNA. The procedure consisted of a pre-read to 60 °C for 30 s, followed by an initial denaturation at 95 °C for 5 min, then 40 cycles (95 °C for 15 s and 60 °C for 60 s), and finally a post-read at 60 °C for 30 s. Afterward, the Hardy-Weinberg equilibrium (HWE) proportion was estimated, and the allelic frequency was consistent with published literature and the NBCI database for this gene (χ^2^(274) = 0.886 and *p* = 0.47). Finally, the genotypes were dummy coded into 0 (no susceptibility allele; homozygosis for the Val allele for the COMT polymorphism) and 1 (at least one susceptibility allele; Val/Met and Met/Met for the COMT).

### 2.3. Procedure

Once the approval of the Ethics Committee was obtained, we contacted several schools to arrange an interview where we explained the objectives and procedure of the investigation to get the authorization of the management team. When we got these informed consents, the same information was transmitted to teachers and families of possible participants during parents’ meetings held in September–October. Subsequently, we gave every family a letter explaining the project in more detail, and together with that, we requested informed consent by explicitly authorizing the participation of their children in the study through the completion of a document. This document was delivered to the school’s tutors in a sealed envelope. The data collection was organized in semesters and performed by adapting to the calendars of the schools that participated in the study and fitting them to their schedules. All the saliva sample collection and most tests were administered from 9.00 to 9.30 in the morning, before the lessons started.

### 2.4. Data Analysis

Before starting with the data analysis, we checked whether the variables followed a normal distribution. Based on these results, ANOVAs and Mann–Whitney analyses were carried out for each of the variables to study possible sex differences. 

Subsequently, due to the existence of sex differences in aggressive behavior, sex-based regression analyses were carried out to analyze the potential moderating role of the COMT polymorphism in the relationship between parenting styles and aggressive behavior. For that, the parenting style variables were separated, creating a “beneficial” (mothers’ and fathers’ authoritative parenting) and a “detrimental” (mothers’ and fathers’ authoritarian parenting) family environment. As the dependent variables did not follow a normal distribution, the regression analyses were therefore conducted using the bootstrapping technique, which performs a bias adjustment and controls for the proportion of type I errors [69]. The associations were considered significant if bootstrapped CIs did not cross zero. Once the regression analyses were performed, to test and identify which theoretical model (diathesis-stress, differential susceptibility, or vantage sensibility) best fits the statistically significant interactions, the techniques described by Roisman et al. [70] were performed. To this end, we used a web-based program developed by Fraley [71]. 

All the analyses, except for the ones conducted in Fraley’s online software (http://www.yourpersonality.net/interaction/, accessed on 24 October 2018), were performed through the statistical package SPSS 25.00 (SPSS Inc., Chicago, IL, USA).

## 3. Results

### 3.1. Analyses of Variance

The results revealed that sex differences exist for proactive aggression (U = 7825.000, *p* = 0.006, and r = 0.134) and reactive aggression (U = 7640.500, *p* = 0.003, and r = 0.178), with boys scoring higher levels than girls in both types of aggressive behaviors. Concerning parenting, there was a statistically significant difference for fathers’ authoritarian parenting style (U = 5709.000, *p* = 0.033, and r = 0.139). There were no statistical differences for the rest of the variables.

### 3.2. The Predictive Role of Temperament, Parenting Styles, and Their Interaction in Relation to Aggression

To examine the predictive role of the COMT polymorphism, mothers’ and fathers’ authoritative parenting styles, and their interactions in boys’ and girls’ aggressive behavior, regression analyses were conducted. As shown in Table 1, the model that tested girls’ proactive aggression (R^2^ = 0.127, F(5, 97) = 2.687, and *p* = 0.026) was statistically significant, with a principal significant effect of mothers’ authoritative parenting style in addition to the interaction “COMT x authoritative mother”.

The model that tested boys’ proactive aggression was also statistically significant (R^2^ = 0.115, F(5, 120) = 2.996, and *p* = 0.014), with the interaction “COMT x authoritative father” statistically significant (Table 1).

Aiming to explore the interaction “COMT x Authoritative Father” found statistically significant in the regression model performed to explore boys’ proactive aggression, we first analyzed the slopes of the regression lines for authoritative father on aggression, separately for the two groups of the COMT variable. The result showed that the relationship was significant for the group with the presence of the Met allele (group 1) (β = −0.51, t(120) = 3.42, and *p* = 0.001), but not for the group that was homozygous for the Val allele (group 0) (β = 0.04, t(120) = 0.19, and *p* = 0.853).

Afterwards, we analyzed at what values of fathers’ authoritative parenting style the relationship between COMT and boys’ proactive aggression was statistically significant. The results revealed that it was statistically significant at low levels (−2 SD in relation to the mean value; β = 1.46, t(120) = 2.68, and *p* = 0.008) but not at high values of the environmental variable (+2 SD in relation to the mean value; β = −0.73, t(120) = 1.17, and *p* = 0.245).

In addition, the Proportion of Interaction (PoI) was explored and a value of 0.20 above the cutoff point was obtained. The PoI is a measure proposed by Roisman et al. [70] to avoid sample size driving the interpretation of the interaction effect, as this metric is unitless and independent of sample size. It is the proportion of the total area between the lines of the interaction plot that is on the positive side (quality of the environment) of the crossover point, and it is calculated by dividing the amount of change “for better” by its sum with the quantity of change “for worse” [72]. PoI values between a 0.40 and 0.60 window indicate an effect consistent with the differential susceptibility model, while the prototypic PoI value for diathesis-stress is 0.00. Therefore, the result was consistent with the diathesis-stress model [70] (Figure 1). 

To study the interaction “COMT x authoritative mother” found significant in the model performed for girls’ proactive aggression, we first analyzed the slopes of the regression lines of mothers’ authoritative parenting style on the aggressive behavior for both groups of COMT. Among those girls with Val/Val alleles of COMT (group 0), the interaction was significant (β = −0.35, t(100) = 3.24, and *p* = 0.002), whereas when they carried the Met allele (group 1), it was not (β = 0.09, t(100) = 1.14, and *p* = 0.259). 

Moreover, we studied at which values of the mothers’ authoritative parenting style the relation between the genetic polymorphism and girls’ proactive aggression was significant. We found out that the relation was statistically significant both at high (+2 SD in relation to the mean value; β = 0.85, t(100) = 2.68, and *p* = 0.009) and low (−2 SD in relation to the mean value; β = −0.93, t(100) = 3.36, and *p* = 0.001) levels of the parenting style. 

Additionally, we obtained a value of 0.46 over the cutoff point when exploring the proportion of interaction (PoI), and it was consistent with the differential susceptibility model [70] (Figure 2). 

To test the predictive value of COMT, mothers’ and fathers’ authoritarian parenting styles and their interactions over girls’ and boys’ reactive and proactive aggression regression analyses were executed. As can be seen in Table 2, the regression models for girls’ reactive (R^2^ = 0.120, F(5, 98) = 2.535, and *p* = 0.034) and proactive aggression (R^2^ = 0.147, F(5, 98) = 3.201, and *p* = 0.010) were statistically significant, with the second one having a statistically significant principal effect of fathers’ authoritarian parenting style and being the interaction “COMT x authoritarian father” statistically significant for this model. Besides, the model performed to test boys’ proactive aggression was also statistically significant (R^2^ = 0.102, F(5, 119) = 2.596, and *p* = 0.029), but there were no statistically significant effects.

To explore the interaction “COMT x authoritarian father” of the regression model performed to analyze girls’ proactive aggressive behavior, the slope of the regression line of fathers’ authoritarian parenting style to girls’ proactive aggression was studied considering the two groups of the COMT variable (group 0 = homozygous for the Val allele and group 1 = presence of the Met allele). The result revealed that the relationship was significant for the group homozygous for the Val allele of COMT (β = 0.50, t(100) = 3.38, and *p* = 0.001), but not for the group with the Met allele of COMT (β = 0.15, t(100) = 1.94, and *p* = 0.056).

Afterwards, we analyzed at what level of fathers’ authoritarian parenting style the relationship between COMT and girls’ proactive aggression was significant. As shown in Figure 3, it was significant at high levels (+2 SD in relation to the mean value; β = −0.76, t(100) = 2.07, and *p* = 0.041) but not at low levels (−2 SD in relation to the mean value; β = 0.65, t(100) = 1.8, and 5, *p* = 0.068) of the parenting style.

Furthermore, the proportion of interaction (PoI) was performed, and a value of 0.57 above the cutoff point was obtained. Considering this information, we could say that the model was consistent with the diathesis-stress model [70] (Figure 3). 

## 4. Discussion

The aim of this study was to examine the predictive significance of the interactions between the Val158Met genetic polymorphism of the COMT gene and parenting styles in relation to aggressive behavior. Specifically, we sought to analyze whether the moderating variable acted as a vulnerability, susceptibility, or vantage factor in the context of parenting styles and its influence on proactive and reactive aggressive behavior in 8-year-old boys and girls.

Regarding proactive aggression in boys, we found that the Met158 allele of the COMT gene was a vulnerability factor to the environment since it predicted higher levels of proactive aggression when boys were exposed to a low democratic upbringing by the father. However, these boys did not benefit from a favorable context, so the result is consistent with the diathesis-stress theory. This finding aligns with the finding of Thompson et al. [72], who found that children homozygous for the Met allele displayed more externalizing behavior in a context characterized by maternal stress. However, there are studies that have linked the Met allele of the COMT gene with susceptibility to the environment and not vulnerability. Zhang et al. [58] found that individuals carrying the Met allele, in addition to being more aggressive in a low-positive parenting context, also benefited from a favorable context showing lower levels of aggression.

According to the tonic–phasic dopamine theory [73,74], the presence of the Met allele, which has a lower dopamine degradation capacity [31], is associated with increased transmission of tonic dopamine (sustained release or sustained levels of dopamine) [75]. This, in turn, results in elevated dopamine levels in the prefrontal cortex [32,33], which has been linked to a higher risk of exhibiting aggressive behaviors [72]. Consequently, individuals with the Met allele have shown greater prefrontal activation and activation in certain regions of the amygdala [76,77]. Williams et al. [78] found that this increased activation may be specific to negative stimuli, suggesting that individuals with the Met allele may have less resilience to negative emotional states and lower emotional control. Furthermore, the brain regions influenced by the Met allele during the processing of these unpleasant stimuli have been found to be involved in aggression [79]. This heightened activation of aggression-related brain areas in response to negative stimuli could explain why our results only showed an association between the Met allele and an adverse context, such as low levels of a democratic style by the father, in explaining the proactive aggressive behavior of boys. In addition to lower emotional resilience to negative states, and according to the warrior/worrier hypothesis, the Met (worried) allele has been associated with better performance on tasks that require behavioral programming, as it confers an advantage in memory and concentration tasks [80,81]. Therefore, the increased aggression displayed by children when exposed to unpleasant contexts could be explained by their reduced resilience to negative states, making them more vulnerable to the negative stimuli of a low-democratic upbringing. At the same time, their enhanced memory and attention skills would enable them to act intentionally and strategically, allowing for “cold” aggressive behaviors such as proactive aggression.

Concerning girls’ aggressive behavior, when exploring the effect of the polymorphism in relation to their proactive aggression, the results revealed that the interaction between the COMT gene and two variables of the family context predicted girls’ proactive aggressive behavior. However, the moderating role of the polymorphism varied depending on the context analyzed. Contrary to the initial hypothesis, the results revealed that girls who were homozygous for the Val allele, in contrast to Met allele carriers, exhibited higher levels of proactive aggression when exposed to an adverse context characterized by a low democratic upbringing by their mother. Conversely, it was also found that girls with the Val/Val genotype displayed lower levels of aggression when exposed to a favorable context, such as a mother with a democratic parenting style. These findings are consistent with the differential susceptibility theory. Similar results were reported by Hygen et al. [56] and Tuvblad et al. [57], who observed that individuals homozygous for the Val allele exhibited higher aggression levels in adverse contexts and lower aggression levels in more favorable contexts when compared to those carrying the Met allele.

However, our results also indicated that girls homozygous for the Val allele were more vulnerable to an authoritarian parenting style from their father compared to individuals carrying the Met allele. In this adverse context, girls with the Val allele exhibited higher levels of proactive aggression. Conversely, they did not benefit from a more favorable context, such as a less authoritarian parental style, which aligns with the diathesis-stress theory. This finding is consistent with Hygen et al. [56], as they reported higher levels of aggressive behavior in individuals homozygous for the Val allele who also had disorganized attachment.

The Val allele is known to have a more efficient dopamine degradation capacity, with a four times higher breakdown capacity compared to the Met allele [31]. According to the tonic–phasic dopamine theory, the Val allele would result in reduced subcortical neurotransmission of tonic dopamine and a decrease in overall dopamine concentration in the prefrontal cortex, while enhancing the transmission of phasic dopamine [73]. Additionally, Perroud et al. [82] suggest that carriers of the valine allele exhibit greater stress-induced phasic dopamine release compared to other genotypes in similar environmental conditions. This explanation supports the observation that valine carriers tend to become more aggressive over time than methionine carriers when facing adversity [56]. Furthermore, according to the warrior/worried hypothesis, the Val (warrior) allele confers resistance to stress, so this genotype would be advantageous in stressful and unpleasant situations [81]. However, due to the tonic reduction in dopamine associated with the Val allele, it has also been linked to lower executive functions and reduced cognitive control [83,84], leading to deficits in response inhibition [85], which would explain the higher levels of aggression observed.

Therefore, the fact that girls homozygous for the Val allele are more vulnerable to an adverse context characterized by authoritarian parenting by the father and more susceptible to democratic parenting by the mother for proactive aggression could be attributed to their advantage in coping with stressful events that would allow them to better face unfavorable situations at first. However, the combination of greater release of phasic dopamine in stressful situations, which can occur in an adverse upbringing, along with less cognitive control and executive functions, eventually leads to the use of aggressive strategies to deal with certain situations. In contrast, in a favorable context, such as a democratic upbringing by the mother, these same girls exhibit lower levels of aggression. This could be due to the fact that democratic parenting has provided them with tools and strategies to cope with deficiencies in cognitive control and executive functions, so that their adaptation strategy will be different in stressful and unpleasant situations where they have an advantage. It is worth noting that this scenario might not occur in a context characterized by a nonauthoritarian father, as his parenting strategy might differ from a democratic one.

In conclusion, our results indicate that the COMT gene polymorphism is a biological factor that confers greater sensitivity to the environment and that, in our study sample, it only interacts with the family environment to predict proactive aggression. This could be explained by the fact that proactive aggression involves a planned attack with the aim of achieving an internal or external reward [60]. Therefore, genes involved in the dopaminergic system, which are relevant to the neural reward system, seem to play a role in the etiology of proactive aggression, as hypothesized by Chen et al. [30]. Furthermore, these results contribute to the debate on whether there is domain-specific or domain-general sensitivity [86]. In line with other research that advocates for domain-specific sensitivity [87,88], the findings of this study suggest that it is domain-specific as the COMT gene only interacts with the environment to predict the proactive form of aggression. Additionally, this research emphasizes the importance of studying the parenting styles of both parents, as the results indicate that both are relevant in predicting children’s aggressive behavior, as found in other studies [49,53], and take sex differences into account. 

Proactive aggression, which is a form of aggression not well understood in terms of its biology, is noteworthy because this study sheds light on it [60]. Furthermore, taking a biosocial perspective on aggressive behavior is crucial for designing future preventive interventions. However, the research has some limitations, which should not be overlooked. First, the results cannot be generalized due to the nonrepresentative nature of the sample, and the nonsignificant findings (effects) may be relevant to the small sample size. Moreover, the family context in this study may not have been highly adverse, and the children’s aggressive behavior exhibited in the study was left-skewed, representing normative levels of aggression. Finally, considering the findings of this study, it would be interesting for future research to measure the effect of epigenetic factors on aggressive behavior since epigenetic changes are influenced by the environment and then exert their influence on gene expression [89].

## Figures and Tables

**Figure 1 brainsci-13-01513-f001:**
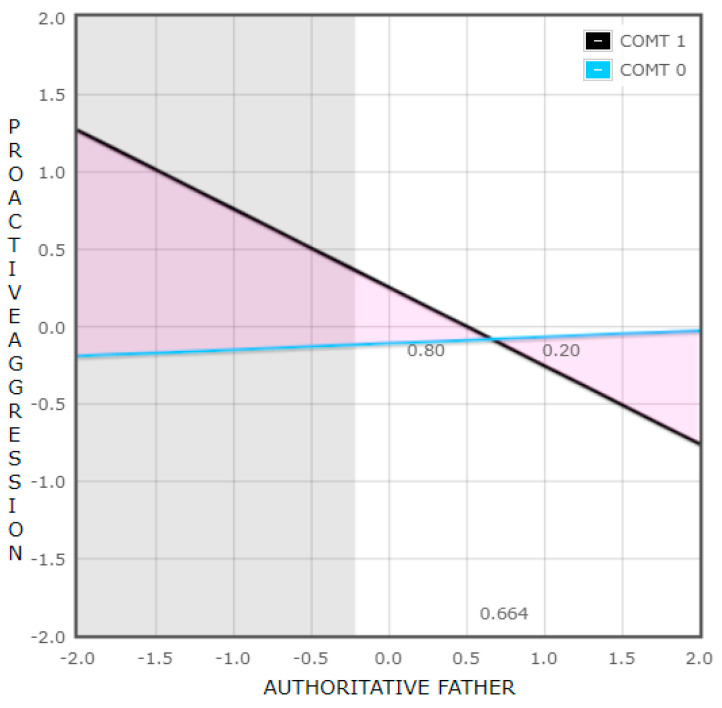
Interaction between COMT and fathers’ authoritative parenting style in relation to proactive aggressive behavior in boys.

**Figure 2 brainsci-13-01513-f002:**
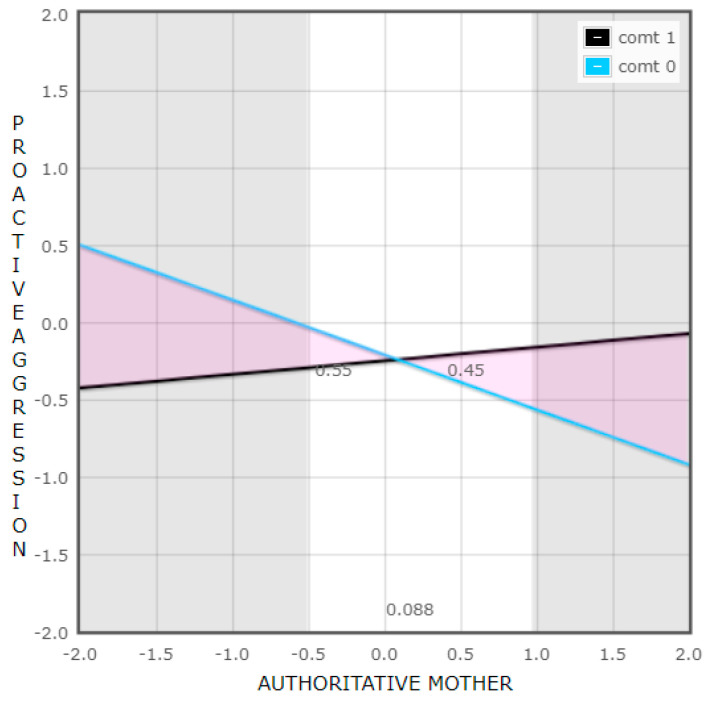
Interaction between COMT and mothers’ authoritative parenting style in relation to proactive aggressive behavior in girls.

**Figure 3 brainsci-13-01513-f003:**
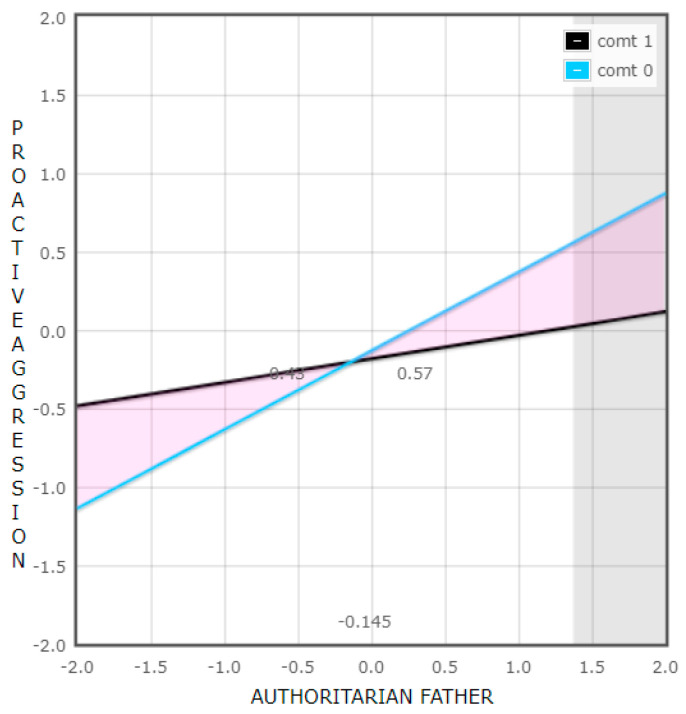
Interaction between COMT and fathers’ authoritarian parenting style in relation to proactive aggressive behavior in girls.

**Table 1 brainsci-13-01513-t001:** Regression analyses for girls’ and boys’ reactive and proactive aggressive behavior, including authoritative mother and authoritative father and COMT.

	Reactive Aggression				Proactive Aggression			
	Girls		Boys		Girls		Boys	
Variable	β	*p*	β	*p*	β	*p*	β	*p*
COMT	−0.127	0.561	0.207	0.283	−0.039	0.714	0.364	0.143
Authoritative mother	−0.262	0.170	0.049	0.789	−0.355	0.007 **	−0.082	0.720
Authoritative father	−0.064	0.712	−0.014	0.927	0.085	0.414	0.041	0.855
COMT x authoritative mother	0.290	0.201	0.080	0.717	0.443	0.002 **	0.333	0.213
COMT x authoritative father	0.033	0.271	−0.127	0.592	−0.097	0.455	−0.548	0.042 *
	R^2^ = 0.037, F(5, 97) = 0.708 *p* = 0.619		R^2^ = 0.025, F(5, 120) = 0.589 *p* = 0.709		R^2^ = 0.127, F(5, 97) = 2.687 *p* = 0.026 *		R^2^ = 0.115, F(5, 120) = 2.996 *p* = 0.014 *	

* *p* < 0.05; ** *p* < 0.01.

**Table 2 brainsci-13-01513-t002:** Regression analyses for girls’ and boys’ reactive and proactive aggressive behavior, including authoritarian mother and authoritarian father and COMT.

	Reactive Aggression				Proactive Aggression			
	Girls		Boys		Girls		Boys	
Variable	β	*p*	β	*p*	β	*p*	β	*p*
COMT	−0.050	0.816	0.160	0.454	−0.051	0.697	0.006	0.103
Authoritarian mother	0.239	0.292	0.360	0.234	−0.089	0.450	−0.023	0.170
Authoritarian father	0.388	0.206	−0.139	0.464	0.502	0.001 ***	0.011	0.232
COMT x authoritarian mother	−0.267	0.270	−0.145	0.685	0.062	0.656	0.009	0.728
COMT x authoritarian father	−0.081	0.791	0.156	0.455	−0.352	0.037 *	−0.009	0.153
	R^2^ = 0.120, F(5, 98) = 2.535 *p* = 0.034 *		R^2^ = 0.07, F(5, 119) = 1.729 *p* = 0.134		R^2^ = 0.147, F(5, 98) = 3.201 *p* = 0.010 **		R^2^ = 0.102, F(5, 119) = 2.596 *p* = 0.029 *	

* *p* < 0.05; ** *p* ≤ 0.01; and *** *p* ≤ 0.001.

## Data Availability

The data presented in this study are available on request from the corresponding author. The data are not publicly available due to the Ethics Committee of the university to which the authors belong.

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
