# Peer review of "The Val158Met Polymorphism in 8-Year-Old Boys and Girls Moderates the Influence of Parenting Styles on Proactive Aggression: Testing the Sensitivity to the Environment"

_brainsci, 2023, doi:10.3390/brainsci13111513_

Round 1

Reviewer 1 Report

Comments and Suggestions for Authors

I would like to say that this study have some strengths, by exploring the parenting and gene interaction on proactive and reactive aggression, and reporting some interesting results. However, I would also suggest that the authors need to make revise regarding the following issues. 1. The major and interesting findings in this study were about Gene * parenting interaction on proactive and reactive aggression. In the INTRODUCTION part, the authors should provide explict descriptions/defination of the the two types of aggression, and also should give clear rationale for exploring gender difference (examined gene * parenting interactions seperately for boys and girls in the RESULT part). 2. Previous studies reported that gene *parenting interaction on reactive aggression among adolescents. Rather this study mainly found G*E interaction on proactive aggression among children.  Given the incondistent results, the authors should give explicit explanation, such that whether the different patterns of G*E interaction were relevant to the different age/developmental stages? 3. I would think that some of the nonsignificant findings(effects) may be relevant to the small sample size. The authors may point this in the discussion part. 4. There were some writing problems and format errors. E.g. (1) 3.2: R2 should be R2 (2 should be superscript). (2) First sentence in 3rd paragraph in 3.2 (and similar sentences in the following paragraphs): confusing sentences. please revise. (3) tonic-phasic thoery of dopamines: please check and keep consistent in expression format. 

Comments on the Quality of English Language

Generally, the writing was good, and the expression was clear. However, there were several confusing sentences, which I have listed in the presious box of comments. Please check and revise. 

Reviewer 2 Report

Comments and Suggestions for Authors

The authors of the manuscript entitled Val158Met polymorphism moderates the effect of parenting styles on proactive aggression in 8-years-old boys and girls: Testing the sensitivity to the environment explored the vulnerability, susceptibility, or vantage sensitivity properties of Val158Met of catechol-O-methyltransferase (COMT) to adverse and favorable parenting styles from both parents in relation to childrens reactive and proactive aggressive behavior. In the present study, 279 eight-year-old children (125 girls and 154 boys) from Spain were recruited and completed the self-reported "Reactive and Proactive Questionnaire" (RPQ) and Parenting Styles and Dimensions Questionnaire (PSDQ). Saliva DNA was used for genotyping via real-time PCR.

The author found that for boys, the Met allele was a vulnerability factor for proactive aggression in response to low authoritative parenting from the father. For girls, it was the Val allele the vulnerability variable to the high authoritarian style of the father and susceptibility factor to the authoritative style of the mother over proactive aggression. The results are discussed attending to possible sex differences. They reached the conclusion that COMT Val158Met polymorphism is a biological variable that confers greater sensitivity to the environment.

I still have concerns about this research.

Major concerns:

  1. Why only 8-year-old boys and girls were recruited in the present study?
  2. The conclusions should be greatly constrained.
  3. Girls carrying the homozygous Val allele were more vulnerable to an authoritarian parenting style from their father, compared to those carrying the Met allele. On the other hand, boys carrying Met158 allele vulnerably had proactive aggression in the context of low authoritative parenting by the father. These data hardly support that COMT plays a role in the etiology of proactive aggression.

Minor concerns:

  1. It would be better that the authors explain the statistics meaning of the Proportion of Interaction (PoI) above the cutoff point was .20. in a figure legend.
  2. The MS needs to be proofread by a native speaker.
Comments on the Quality of English Language

The English writing can be improved. The MS is too wordy, such as 'However, the differences were more pronounced for those carrying the Met allele. However, this research is not the only one in which both the Val and the Met alleles have been found to confer a sensitivity to the environment.' 

Reviewer 3 Report

Comments and Suggestions for Authors I have a few comments: - In the title, it is not clear whether the Val158Met polymorphism was assessed in the parents or in the children - Introduction - the first sentence: Aggressive children may also develop criminal behavior in the adulthood, leading to imprisonment. - It is not probable that aggressive behavior of children is mainly influenced by the Val158Met polymorphism, in psychiatry, every trait is polygenic and with a complex etiology - Statistics: Was the number of study subjects based on a statistical power analysis? Was Bonferroni correction applied or not, and why? - It is nice that the authors found a genetic association with behavior, but epigenetics may also play an important role, and this was not assesseed in the study - Were testosteron and estrogen important in the children´s aggressive behavior, or is it not relevant in 8-year old children? (Btw., estrogen also influences the COMT gene expression.)

Round 2

Reviewer 2 Report

Comments and Suggestions for Authors

I do not have further concerns.

Comments on the Quality of English Language

The quality of English language is good.

Author Response

Minor editing of English language made.